# Node Dependent Local Smoothing for Scalable Graph Learning

**Wentao Zhang[1], Mingyu Yang[1], Zeang Sheng[1], Yang Li[1]**
**Wen Ouyang[2], Yangyu Tao[2], Zhi Yang[1,3], Bin Cui[1,3,4]**
[1]School of CS, Peking University [2]Tencent Inc.
[3] Key Lab of High Confidence Software Technologies, Peking University
[4]Institute of Computational Social Science, Peking University (Qingdao), China
[1]{wentao.zhang, ymyu, shengzeang18, liyang.cs, yangzhi, bin.cui}@pku.edu.cn
[2]{gdpouyang, brucetao}@tencent.com

## Abstract

Recent works reveal that feature or label smoothing lies at the core of Graph Neural Networks (GNNs). Concretely, they show feature smoothing combined with simple linear regression achieves comparable performance with the carefully designed GNNs, and a simple MLP model with label smoothing of its prediction can outperform the vanilla GCN. Though an interesting finding, smoothing has not been well understood, especially regarding how to control the extent of smoothness. Intuitively, too small or too large smoothing iterations may cause *under-smoothing* or *over-smoothing* and can lead to sub-optimal performance. Moreover, the extent of smoothness is node-specific, depending on its degree and local structure. To this end, we propose a novel algorithm called node-dependent local smoothing (NDLS), which aims to control the smoothness of every node by setting a node-specific smoothing iteration. Specifically, NDLS computes influence scores based on the adjacency matrix and selects the iteration number by setting a threshold on the scores. Once selected, the iteration number can be applied to both feature smoothing and label smoothing. Experimental results demonstrate that NDLS enjoys high accuracy – state-of-the-art performance on node classifications tasks, flexibility – can be incorporated with any models, scalability and efficiency – can support large scale graphs with fast training.

## 1 Introduction

In recent years, Graph Neural Networks (GNNs) have received a surge of interest with the state-of-the-art performance on many graph-based tasks [2, 36, 9, 34, 28, 29]. Recent works have found that the success of GNNs can be mainly attributed to smoothing, either at feature or label level. For example, SGC [27] shows using smoothed features as input to a simple linear regression model achieves comparable performance with lots of carefully designed and complex GNNs. At the smoothing stage, features of neighbor nodes are aggregated and combined with the current node's feature to form smoothed features. This process is often iterated multiple times. The smoothing is based on the assumption that labels of nodes that are close to each other are highly correlated, therefore, the features of nodes nearby should help predict the current node's label.

One crucial and interesting parameter of neighborhood feature aggregation is the number of smoothing iterations $k$, which controls how much information is being gathered. Intuitively, an aggregation process of $k$ iterations (or layers) enables a node to leverage information from nodes that are $k$-hop away [22, 33]. The choice of $k$ is closely related to the structural properties of graphs and has a

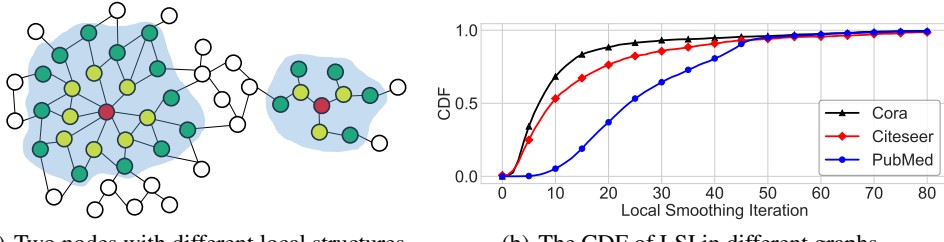

(a) Two nodes with different local structures      (b) The CDF of LSI in different graphs

Figure 1: (Left) The node in dense region has larger smoothed area within two iterations of propagation. (Right) The CDF of LSI in three citation networks.

significant impact on the model performance. However, most existing GNNs only consider the fixed-length propagation paradigm – a uniform $k$ for all the nodes. This is problematic since the number of iterations should be *node dependent* based on its degree and local structures. For example, as shown in Figure 1(a), the two nodes have rather different local structures, with the left red one resides in the center of a dense cluster and the right red one on the periphery with few connections. The number of iterations to reach an optimal level of smoothness are rather different for the two nodes. Ideally, poorly connected nodes (e.g., the red node on the right) needs large iteration numbers to efficiently gather information from other nodes while well-connected nodes (e.g., the red node on the left) should keep the iteration number small to avoid *over-smoothing*. Though some learning-based approaches have proposed to adaptively aggregate information for each node through gate/attention mechanism or reinforcement learning [25, 17, 35, 23], the performance gains are at the cost of increased training complexity, hence not suitable for scalable graph learning.

In this paper, we propose a simple yet effective solution to this problem. Our approach, called node-dependent local smoothing (NDLS), calculates a node-specific iteration number for each node, referred to as local smooth iteration (LSI). Once the LSI for a specific node is computed, the corresponding local smoothing algorithm only aggregates the information from the nodes within a distance less than its LSI as the new feature. The LSI is selected based on influence scores, which measure how other nodes influence the current node. NDLS sets the LSI for a specific node to be the minimum number of iterations so that the influence score is $\epsilon$-away from the *over-smoothing* score, defined as the influence score at infinite iteration. The insight is that each node's influence score should be at a reasonable level. Since the nodes with different local structures have different "smoothing speed", we expect the iteration number to be adaptive. Figure 1(b) illustrates Cumulative Distribution Function (CDF) for the LSI of individual nodes in real-world graphs. The heterogeneous and long-tail property exists in all the datasets, which resembles the characteristics of the degree distribution of nodes in real graphs.

Based on NDLS, we propose a new graph learning algorithm with three stages: (1) feature smoothing with NDLS (NDLS-F); (2) model training with smoothed features; (3) label smoothing with NDLS (NDLS-L). Note that in our framework, the graph structure information is only used in pre-processing and post-processing steps, i.e., stages (1) and (3) (See Figure 2). Our NDLS turns a graph learning problem into a vanilla machine learning problem with independent samples. This simplicity enables us to train models on larger-scale graphs. Moreover, our NDLS kernel can act as a drop-in replacement for any other graph kernels and be combined with existing models such as Multilayer Perceptron (MLP), SGC [27], SIGN [24], $S^2$GC [37] and GBP [5].

Extensive evaluations on seven benchmark datasets, including large-scale datasets like ogbn-papers100M [13], demonstrates that NDLS achieves not only the state-of-the-art node classification performance but also high training scalability and efficiency. Especially, NDLS outperforms APPNP [25] and GAT [26] by a margin of $1.0\%$-$1.9\%$ and $0.9\%$-$2.4\%$ in terms of test accuracy, while achieving up to $39\times$ and $186\times$ training speedups, respectively.

## 2 Preliminaries

In this section, we first introduce the semi-supervised node classification task and review the prior models, based on which we derive our method in Section 3. Consider a graph $\mathcal{G} = (\mathcal{V}, \mathcal{E})$ with $|\mathcal{V}| = n$

nodes and $|\mathcal{E}| = m$ edges, the adjacency matrix (including self loops) is denoted as $\tilde{\mathbf{A}} \in \mathbb{R}^{n \times n}$ and the feature matrix is denoted as $\mathbf{X} = \{\boldsymbol{x}_1, \boldsymbol{x}_2..., \boldsymbol{x}_n\}$ in which $\boldsymbol{x}_i \in \mathbb{R}^f$ represents the feature vector of node $v_i$. Besides, $\mathbf{Y} = \{\boldsymbol{y_1}, \boldsymbol{y_2}..., \boldsymbol{y_l}\}$ is the initial label matrix consisting of one-hot label indicator vectors. The goal is to predict the labels for nodes in the unlabeled set $\mathcal{V}_u$ with the supervision of labeled set $\mathcal{V}_l$.

**GCN** smooths the representation of each node via aggregating its own representations and the ones of its neighbors'. This process can be defined as

$$\mathbf{X}^{(k+1)} = \delta \left( \hat{\mathbf{A}} \mathbf{X}^{(k)} \mathbf{W}^{(k)} \right), \qquad \hat{\mathbf{A}} = \widetilde{\mathbf{D}}^{r-1} \tilde{\mathbf{A}} \widetilde{\mathbf{D}}^{-r}, \tag{1}$$

where $\hat{\mathbf{A}}$ is the normalized adjacency matrix, $r \in [0, 1]$ is the convolution coefficient, and $\widetilde{\mathbf{D}}$ is the diagonal node degree matrix with self loops. Here $\mathbf{X}^{(k)}$ and $\mathbf{X}^{(k+1)}$ are the smoothed node features of layer $k$ and $k + 1$ respectively while $\mathbf{X}^{(0)}$ is set to $\mathbf{X}$, the original feature matrix. In addition, $\mathbf{W}^{(k)}$ is a layer-specific trainable weight matrix at layer $k$, and $\delta(\cdot)$ is the activation function. By setting $r = 0.5, 1$ and $0$, the convolution matrix $\widetilde{\mathbf{D}}^{r-1} \tilde{\mathbf{A}} \widetilde{\mathbf{D}}^{-r}$ represents the symmetric normalization adjacency matrix $\widetilde{\mathbf{D}}^{-1/2} \tilde{\mathbf{A}} \widetilde{\mathbf{D}}^{-1/2}$ [16], the transition probability matrix $\tilde{\mathbf{A}} \widetilde{\mathbf{D}}^{-1}$ [32], and the reverse transition probability matrix $\widetilde{\mathbf{D}}^{-1} \tilde{\mathbf{A}}$ [30], respectively.

**SGC.** For each GCN layer defined in Eq. 1, if the non-linear activation function $\delta(\cdot)$ is an identity function and $\mathbf{W}^{(k)}$ is an identity matrix, we get the smoothed feature after $k$-iterations propagation as $\mathbf{X}^{(k)} = \hat{\mathbf{A}}^k \mathbf{X}$. Recent studies have observed that GNNs primarily derive their benefits from performing feature smoothing over graph neighborhoods rather than learning non-linear hierarchies of features as implied by the analogy to CNNs [21, 8, 12]. By hypothesizing that the non-linear transformations between GCN layers are not critical, SGC [27] first extracts the smoothed features $\mathbf{X}^{(k)}$ then feeds them to a linear model, leading to higher scalability and efficiency. Following the design principle of SGC, piles of works have been proposed to further improve the performance of SGC while maintaining high scalability and efficiency, such as SIGN [24], S$^2$GC [37] and GBP [5].

**Over-Smoothing [18] issue.** By continually smoothing the node feature with infinite number of propagation in SGC, the final smoothed feature $\mathbf{X}^{(\infty)}$ is

$$\mathbf{X}^{(\infty)} = \hat{\mathbf{A}}^\infty \mathbf{X}, \qquad \hat{\mathbf{A}}_{i,j}^\infty = \frac{(d_i + 1)^r (d_j + 1)^{1-r}}{2m + n}, \tag{2}$$

where $\hat{\mathbf{A}}^\infty$ is the final smoothed adjacency matrix, $\hat{\mathbf{A}}_{i,j}^\infty$ is the weight between nodes $v_i$ and $v_j$, $d_i$ and $d_j$ are the node degrees for $v_i$ and $v_j$, respectively. Eq. (2) shows that as we smooth the node feature with an infinite number of propagations in SGC, the final feature is over-smoothed and unable to capture the full graph structure information since it only relates with the node degrees of target nodes and source nodes. For example, if we set $r = 0$ or $1$, all nodes will have the same smoothed features because only the degrees of the source or target nodes have been considered.

## 3  Local Smoothing Iteration (LSI)

The features after $k$ iterations of smoothing is $\mathbf{X}^{(k)} = \hat{\mathbf{A}}^k \mathbf{X}$. Inspired by [30], we measure the influence of node $v_j$ on node $v_i$ by measuring how much a change in the input feature of $v_j$ affects the representation of $v_i$ after $k$ iterations. For any node $v_i$, the influence vector captures the influences of all other nodes. Considering the $h^{th}$ feature of $\mathbf{X}$, we define an influence matrix $I_h(k)$:

$$I_h(k)_{ij} = \frac{\partial \hat{\mathbf{X}}_{ih}^{(k)}}{\partial \hat{\mathbf{X}}_{jh}^{(0)}}. \tag{3}$$

$$I(k) = \hat{\mathbf{A}}^k, \tilde{I}_i = \hat{\mathbf{A}}^\infty \tag{4}$$

Since $I_h(k)$ is independent to $h$, we replace $I_h(k)$ with $I(k)$, which can be further represented as $I(k) = I_h(k), \ \forall h \in \{1, 2, .., f\}$, where $f$ indicates the number of features of $\mathbf{X}$. We denote $I(k)_i$ as the $i^{th}$ row of $I(k)$, and $\tilde{I}$ as $I(\infty)$. Given the normalized adjacency matrix $\hat{\mathbf{A}}$, we can

have $I(k) = \hat{\mathbf{A}}^k$ and $\tilde{I} = \hat{\mathbf{A}}^\infty$. According to Eq. (2), $\tilde{I}$ converges to a unique stationary matrix independent of the distance between nodes, resulting in that the aggregated features of nodes are merely relative with their degrees (i.e., over-smoothing).

We denote $I(k)_i$ as the $i^{th}$ row of $I(k)$, and it means the influence from the other nodes to the node $v_i$ after $k$ iterations of propagation. We introduce a new concept *local smoothing iteration* (parameterized by $\epsilon$), which measures the minimal number of iterations $k$ required for the influence of other nodes on node $v_i$ to be within an $\epsilon$-distance to the over-smoothing stationarity $\tilde{I}_i$.

**Definition 3.1. Local-Smoothing Iteration** *(LSI, parameterized by $\epsilon$) is defined as*

$$K(i, \epsilon) = \min\{k : ||\tilde{I}_i - I(k)_i||_2 < \epsilon\}, \tag{5}$$

*where $|| \cdot ||_2$ is two-norm, and $\epsilon$ is an arbitrary small constant with $\epsilon > 0$.*

Here $\epsilon$ is a graph-specific parameter, and a smaller $\epsilon$ indicates a stronger smoothing effect. The $\epsilon$-distance to the over-smoothing stationarity $\tilde{I}_i$ ensures that the smooth effect on node $v_i$ is sufficient and bounded to avoid over-smoothing. As shown in Figure 1(b), we can have that the distribution of LSI owns the *heterogeneous and long-tail property*, where a large percentage of nodes have much smaller LSI than the rest. Therefore, the required LSI to approach the stationarity is heterogeneous across nodes. Now we discuss the connection between LSI and node local structure, showcasing nodes in the sparse region (e.g., both the degrees of itself and its neighborhood are low) can greatly prolong the iteration to approach over-smoothing stationarity. This heterogeneity property is not fully utilized in the design of current GNNs, leaving the model design in a dilemma between unnecessary iterations for a majority of nodes and insufficient iterations for the rest of nodes. Hence, by adaptively choosing the iteration based on LSI for different nodes, we can significantly improve model performance.

**Theoretical Properties of LSI.** We now analyze the factors determining the LSI of a specific node. To facilitate the analysis, we set the coefficient $r = 0$ for the normalized adjacency matrix $\hat{\mathbf{A}}$ in Eq. (1), thus $\hat{\mathbf{A}} = \tilde{\mathbf{D}}^{-1}\tilde{\mathbf{A}}$. The proofs of following theorems can be found in Appendix A.1.

**Theorem 3.1.** *Given feature smoothing $\mathbf{X}^{(k)} = \hat{\mathbf{A}}^k\mathbf{X}$ with $\hat{\mathbf{A}} = \tilde{\mathbf{D}}^{-1}\tilde{\mathbf{A}}$, we have*

$$K(i, \epsilon) \leq \log_{\lambda_2}\left(\epsilon\sqrt{\frac{\tilde{d}_i}{2m + n}}\right), \tag{6}$$

*where $\lambda_2$ is the second largest eigenvalue of $\hat{\mathbf{A}}$, $\tilde{d}_i$ denotes the degree of node $v_i$ plus 1 (i.e., $\tilde{d}_i = d_i + 1$), and $m$, $n$ denote the number of edges and nodes respectively.*

Note that $\lambda_2 \leq 1$. Theorem 3.1 shows that the upper-bound of the LSI is positively correlated with the scale of the graph $(m, n)$, the sparsity of the graph (small $\lambda_2$ means strong connection and low sparsity, and vice versa), and negatively correlated with the degree of node $v_i$.

**Theorem 3.2.** *For any nodes $i$ in a graph $\mathcal{G}$,*

$$K(i, \epsilon) \leq \max\{K(j, \epsilon), j \in N(i)\} + 1, \tag{7}$$

*where $N(i)$ is the set of node $v_i$'s neighbours.*

Theorem 3.2 indicates that the difference between two neighboring nodes' LSIs is no more than 1, therefore the nodes with a super-node as neighbors (or neighbor's neighbors) may have small LSIs. That is to say, the sparsity of the local area, where a node locates, also affects its LSI positively. Considering Theorems 3.1 and 3.2 together, we can have a union upper-bound of $K(i, \epsilon)$ as

$$K(i, \epsilon) \leq \min\left\{\max\{K(j, \epsilon), j \in N(i)\} + 1, \log_{\lambda_2}\left(\epsilon\sqrt{\frac{\tilde{d}_i}{2m + n}}\right)\right\}. \tag{8}$$

## 4  NDLS Pipeline

The basic idea of NDLS is to utilize the LSI heterogeneity to perform a node-dependent aggregation over a neighborhood within a distance less than the specific LSI for each node. Further, we propose

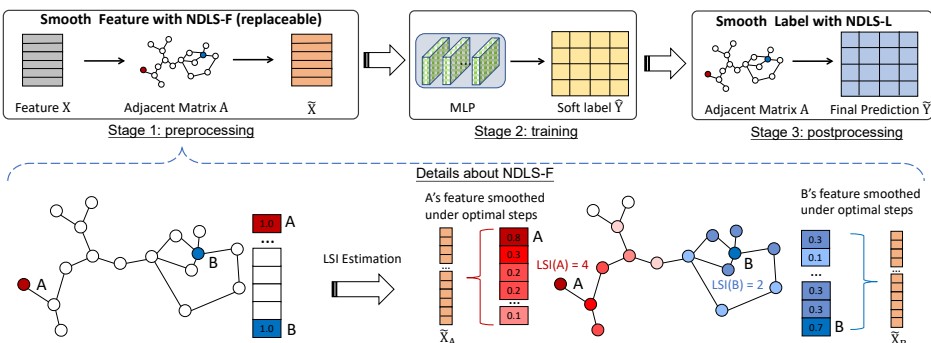

Figure 2: Overview of the proposed NDLS method, including (1) feature smoothing with NDLS (NDLS-F), (2) model training with smoothed features, and (3) label smoothing with NDLS (NDLS-L). NDLS-F and NDLS-L correspond to pre-processing and post-processing steps respectively.

a simple pipeline with three main parts (See Figure 2): (1) a node-dependent local smoothing of the feature (NDLS-F) over the graph, (2) a base prediction result with the smoothed feature, (3) a node-dependent local smoothing of the label predictions (NDLS-L) over the graph. Note this pipeline is not trained in an end-to-end way, the stages (1) and (3) in NDLS are only the pre-processing and post-processing steps, respectively. Furthermore, the graph structure is only used in the pre/post-processing NDLS steps, not for the base predictions. Compared with prior GNN models, this key design enables higher scalability and a faster training process.

Based on the graph structure, we first compute the node-dependent *local smoothing iteration* that maintains a proper distance to the over-smoothing stationarity. Then the corresponding local smoothing kernel only aggregates the information (feature or prediction) for each node from the nodes within a distance less than its LSI value. The combination of NDLS-F and NDLS-L takes advantage of both label smoothing (which tends to perform fairly well on its own without node features) and the node feature smoothing. We will see that combining these complementary signals yields state-of-the-art predictive accuracy. Moreover, our NDLS-F kernel can act as a drop-in replacement for graph kernels in other scalable GNNs such as SGC, $S^2$GC, GBP, etc.

## 4.1 Smooth Features with NDLS-F

Once the node-dependent LSI $K(i, \epsilon)$ for a specific node $i$ is obtained, we smooth the initial input feature $\mathbf{X}_i$ of node $i$ with node-dependent LSI as:

$$\widetilde{\mathbf{X}}_i(\epsilon) = \frac{1}{K(i,\epsilon)+1} \sum_{k=0}^{K(i,\epsilon)} \mathbf{X}_i^{(k)}. \tag{9}$$

To capture sufficient neighborhood information, for each node $v_i$, we average its multi-scale features $\{\mathbf{X}_i^{(k)} \mid k \leq K(i,\epsilon)\}$ obtained by aggregating information within $k$ hops from the node $v_i$.

The matrix form of the above equation can be formulated as

$$\widetilde{\mathbf{X}}(\epsilon) = \sum_{k=0}^{\max_i K(i,\epsilon)} \mathbf{M}^{(k)}\mathbf{X}^{(k)}, \qquad \mathbf{M}^{(\mathbf{k})}{}_{ij} = \begin{cases} \frac{1}{K(i,\epsilon)+1}, & i = j \quad and \quad k \leq K(i,\epsilon) \\ 0, & \text{otherwise} \end{cases}, \tag{10}$$

where $\mathbf{M}^{(\mathbf{k})}$ is a set of diagonal matrix.

## 4.2 Simple Base Prediction

With the smoothed feature $\widetilde{\mathbf{X}}$ according to Eq. 9, we then train a model to minimize the loss – $\sum_{v_i \in \mathcal{V}_l} \ell\left(\boldsymbol{y}_i, f(\widetilde{\mathbf{X}}_i)\right)$, where $\widetilde{\mathbf{X}}_i$ denotes the $i^{th}$ row of $\widetilde{\mathbf{X}}$, $\ell$ is the cross-entropy loss function, and $f(\widetilde{\mathbf{X}}_i)$ is the predictive label distribution for node $v_i$. In NDLS, the default $f$ is a MLP model and

$\hat{\mathbf{Y}} = f(\widetilde{\mathbf{X}})$ is its soft label predicted (softmax output). Note that, many other models such as Random Forest [20] and XGBoost [6] could also be used in NDLS (See more results in Appendix A.2).

### 4.3 Smooth Labels with NDLS-L

Similar to the feature propagation, we can also propagate the soft label $\hat{\mathbf{Y}}$ with $\hat{\mathbf{Y}}^{(k)} = \hat{\mathbf{A}}^k \hat{\mathbf{Y}}$. Considering the influence matrix of softmax label $J_h(k)$.

$$J_h(k)_{ij} = \frac{\partial \hat{\mathbf{Y}}_{ih}^{(k)}}{\partial \hat{\mathbf{Y}}_{jh}^{(0)}}. \tag{11}$$

According to the definition above we have that

$$J_h(k) = I_h(k), \forall h \in \{1, 2, .., f\}. \tag{12}$$

Therefore, local smoothing can be further applied to address over-smoothing in label propagation. Concretely, we smooth an initial soft label $\hat{\mathbf{Y}}_i$ of node $v_i$ with NDLS as follows

$$\widetilde{\mathbf{Y}}_i(\epsilon) = \frac{1}{K(i,\epsilon) + 1} \sum_{k=0}^{K(i,\epsilon)} \hat{\mathbf{Y}}_i^{(k)}. \tag{13}$$

Similarly, the matrix form of the above equation can be formulated as

$$\widetilde{\mathbf{Y}}(\epsilon) = \sum_{k=0}^{\max_i K(i,\epsilon)} \mathbf{M}^{(k)} \hat{\mathbf{Y}}^{(k)}, \tag{14}$$

where $\mathbf{M}^{(\mathbf{k})}$ follows the definition in Eq. (10).

## 5 Comparison with Existing Methods

**Decoupled GNNs.** The aggregation and transformation operations in coupled GNNs (i.e., GCN [15], GAT [26] and JK-Net [30]) are inherently intertwined in Eq. (1), so the propagation iterations $L$ always equals to the transformation iterations $K$. Recently, some decoupled GNNs (e.g., PPNP [16], PPRGo [1], APPNP [16], AP-GCN [25] and DAGNN [21]) argue the entanglement of these two operations limits the propagation depth and representation ability of GNNs, so they first do the transformation and then smooth and propagate the predictive soft label with higher depth in an end-to-end manner. Especially, AP-GCN and DAGNN both use a learning mechanism to learn propagation adaptively. Unfortunately, all these coupled and decoupled GNNs are hard to scale to large graphs – *scalability issue* since they need to repeatedly perform an expensive recursive neighborhood expansion in multiple propagations of the features or soft label predicted. NDLS addresses this issue by dividing the training process into multiple stages.

**Sampling-based GNNs.** An intuitive method to tackle the recursive neighborhood expansion problem is sampling. As a node-wise sampling method, GraphSAGE [11] samples the target nodes as a mini-batch and samples a fixed size set of neighbors for computing. VR-GCN [4] analyzes the variance reduction on node-wise sampling, and it can reduce the size of samples with an additional memory cost. In the layer level, Fast-GCN [3] samples a fixed number of nodes at each layer, and ASGCN [14] proposes the adaptive layer-wise sampling with better variance control. For the graph-wise sampling, Cluster-GCN [7] clusters the nodes and only samples the nodes in the clusters, and GraphSAINT [32] directly samples a subgraph for mini-batch training. We don't use sampling in NDLS since the sampling quality highly influences the classification performance.

**Linear Models.** Following SGC [27], some recent methods remove the non-linearity between each layer in the forward propagation. SIGN [24] allows using different local graph operators and proposes to concatenate the different iterations of propagated features. $S^2GC$ [37] proposes the simple spectral graph convolution to average the propagated features in different iterations. In addition, GBP [5] further improves the combination process by weighted averaging, and all nodes in the same layer share the same weight. In this way, GBP considers the smoothness in a layer perspective way. Similar

Table 1: Algorithm analysis for existing scalable GNNs. $n$, $m$, $c$, and $f$ are the number of nodes, edges, classes, and feature dimensions, respectively. $b$ is the batch size, and $k$ refers to the number of sampled nodes. $L$ corresponds to the number of times we aggregate features, $K$ is the number of layers in MLP classifiers. For the coupled GNNs, we always have $K = L$.

| Type | Method | Preprocessing and postprocessing | Training | Inference | Memory |
|------|--------|----------------------------------|----------|-----------|--------|
| Node-wise sampling | GraphSAGE | - | $\mathcal{O}(k^L n f^2)$ | $\mathcal{O}(k^L n f^2)$ | $\mathcal{O}(b k^L f + L f^2)$ |
| Layer-wise sampling | FastGCN | - | $\mathcal{O}(k L n f^2)$ | $\mathcal{O}(k L n f^2)$ | $\mathcal{O}(b k L f + L f^2)$ |
| Graph-wise sampling | Cluster-GCN | $\mathcal{O}(m)$ | $\mathcal{O}(L m f + L n f^2)$ | $\mathcal{O}(L m f + L n f^2)$ | $\mathcal{O}(b L f + L f^2)$ |
| Linear model | SGC | $\mathcal{O}(L m f)$ | $\mathcal{O}(n f^2)$ | $\mathcal{O}(n f^2)$ | $\mathcal{O}(b f + f^2)$ |
| | $S^2$GC | $\mathcal{O}(L m f)$ | $\mathcal{O}(n f^2)$ | $\mathcal{O}(n f^2)$ | $\mathcal{O}(b f + f^2)$ |
| | SIGN | $\mathcal{O}(L m f)$ | $\mathcal{O}(K n f^2)$ | $\mathcal{O}(K n f^2)$ | $\mathcal{O}(b L f + K f^2)$ |
| | GBP | $\mathcal{O}(L n f + L \frac{\sqrt{m \lg n}}{\varepsilon})$ | $\mathcal{O}(K n f^2)$ | $\mathcal{O}(K n f^2)$ | $\mathcal{O}(b f + K f^2)$ |
| Linear model | NDLS | $\mathcal{O}(L m f + L m c)$ | $\mathcal{O}(K n f^2)$ | $\mathcal{O}(K n f^2)$ | $\mathcal{O}(b f + K f^2)$ |

Table 2: Overview of datasets and task types (T/I represents Transductive/Inductive).

| Dataset | #Nodes | #Features | #Edges | #Classes | #Train/Val/Test | Type | Description |
|---------|--------|-----------|--------|----------|-----------------|------|-------------|
| Cora | 2,708 | 1,433 | 5,429 | 7 | 140/500/1,000 | T | citation network |
| Citeseer | 3,327 | 3,703 | 4,732 | 6 | 120/500/1,000 | T | citation network |
| Pubmed | 19,717 | 500 | 44,338 | 3 | 60/500/1,000 | T | citation network |
| Industry | 1,000,000 | 64 | 1,434,382 | 253 | 5K/10K/30K | T | short-form video network |
| ogbn-papers100M | 111,059,956 | 128 | 1,615,685,872 | 172 | 1,207K/125K/214K | T | citation network |
| Flickr | 89,250 | 500 | 899,756 | 7 | 44K/22K/22K | I | image network |
| Reddit | 232,965 | 602 | 11,606,919 | 41 | 155K/23K/54K | I | social network |

to these works, we also use a linear model for higher training scalability. The difference lies in that we consider the smoothness from a node-dependent perspective and each node in NDLS has a personalized aggregation iteration with the proposed local smoothing mechanism.

Table 1 compares the asymptotic complexity of NDLS with several representative and scalable GNNs. In the stage of the preprocessing, the time cost of clustering in Cluster-GCN is $\mathcal{O}(m)$ and the time complexity of most linear models is $\mathcal{O}(L m f)$. Besides, NDLS has an extra time cost $\mathcal{O}(L m c)$ for the postprocessing in label smoothing. GBP conducts this process approximately with a bound of $\mathcal{O}(L n f + L \frac{\sqrt{m \lg n}}{\varepsilon})$, where $\varepsilon$ is a error threshold. Compared with the sampling-based GNNs, the linear models usually have smaller training and inference complexity, i.e., higher efficiency. Memory complexity is a crucial factor in large-scale graph learning because it is difficult for memory-intensive algorithms such as GCN and GAT to train large graphs on a single machine. Compared with SIGN, both GBP and NDLS do not need to store smoothed features in different iterations, and the feature storage complexity can be reduced from $\mathcal{O}(b L f)$ to $\mathcal{O}(b f)$.

# 6 Experiments

In this section, we verify the effectiveness of NDLS on seven real-world graph datasets. We aim to answer the following four questions. **Q1:** Compared with current SOTA GNNs, can NDLS achieve higher predictive accuracy and why? **Q2:** Are NDLS-F and NDLS-L better than the current feature and label smoothing mechanisms (e.g., the weighted feature smoothing in GBP and the adaptive label smoothing in DAGNN)? **Q3:** Can NDLS obtain higher efficiency over the considered GNN models? **Q4:** How does NDLS perform on sparse graphs (i.e., low label/edge rate, missing features)?

## 6.1 Experimental Setup

**Datasets.** We conduct the experiments on (1) six publicly partitioned datasets, including four citation networks (Citeseer, Cora, PubMed, and ogbn-papers100M) in [15, 13] and two social networks (Flickr and Reddit) in [32], and (2) one short-form video recommendation graph (Industry) from our industrial cooperative enterprise. The dataset statistics are shown in Table 2 and more details about these datasets can be found in Appendix A.3.

**Baselines.** In the transductive setting, we compare our method with (1) the coupled GNNs: GCN [15], GAT [26] and JK-Net [30]; (2) the decoupled GNNs: APPNP [16], AP-GCN [25],

Table 3: Results of transductive settings. OOM means "out of memory".

| Type | Models | Cora | Citeseer | PubMed | Industry | ogbn-papers100M |
|------|--------|------|----------|--------|----------|-----------------|
| Coupled | GCN | 81.8±0.5 | 70.8±0.5 | 79.3±0.7 | 45.9±0.4 | OOM |
| | GAT | 83.0±0.7 | 72.5±0.7 | 79.0±0.3 | 46.8±0.7 | OOM |
| | JK-Net | 81.8±0.5 | 70.7±0.7 | 78.8±0.7 | 47.2±0.3 | OOM |
| Decoupled | APPNP | 83.3±0.5 | 71.8±0.5 | 80.1±0.2 | 46.7±0.6 | OOM |
| | AP-GCN | 83.4±0.3 | 71.3±0.5 | 79.7±0.3 | 46.9±0.7 | OOM |
| | PPRGo | 82.4±0.2 | 71.3±0.5 | 80.0±0.4 | 46.6±0.5 | OOM |
| | DAGNN (Gate) | 84.4±0.5 | 73.3±0.6 | 80.5±0.5 | 47.1±0.6 | OOM |
| | DAGNN (NDLS-L)* | 84.4±0.6 | 73.6±0.7 | 80.9±0.5 | 47.2±0.7 | OOM |
| Linear | MLP | 61.1±0.6 | 61.8±0.8 | 72.7±0.6 | 41.3±0.8 | 47.2±0.3 |
| | SGC | 81.0±0.2 | 71.3±0.5 | 78.9±0.5 | 45.2±0.3 | 63.2±0.2 |
| | SIGN | 82.1±0.3 | 72.4±0.8 | 79.5±0.5 | 46.3±0.5 | 64.2±0.2 |
| | $S^2$GC | 82.7±0.3 | 73.0±0.2 | 79.9±0.3 | 46.6±0.6 | 64.7±0.3 |
| | GBP | 83.9±0.7 | 72.9±0.5 | 80.6±0.4 | 46.9±0.7 | 65.2±0.3 |
| Linear | NDLS-F+MLP* | 84.1±0.6 | 73.5±0.5 | 81.1±0.6 | 47.5±0.7 | 65.3±0.5 |
| | MLP+NDLS-L* | 83.9±0.6 | 73.1±0.8 | 81.1±0.6 | 46.9±0.7 | 64.6±0.4 |
| | SGC+NDLS-L* | 84.2±0.2 | 73.4±0.5 | 81.1±0.4 | 47.1±0.6 | 64.9±0.3 |
| | NDLS* | **84.6±0.5** | **73.7±0.6** | **81.4±0.4** | **47.7±0.5** | **65.6±0.3** |

DAGNN (Gate) [21], and PPRGo [1]; (3) the linear-model-based GNNs: MLP, SGC [27], SIGN [24], $S^2$GC [37] and GBP [5]. In the inductive setting, the compared baselines are sampling-based GNNs: GraphSAGE [11], FastGCN [3], ClusterGCN [7] and GraphSAINT [32]. Detailed descriptions of these baselines are provided in Appendix A.4.

**Implementations.** To alleviate the influence of randomness, we repeat each method ten times and report the mean performance. The hyper-parameters of baselines are tuned by OpenBox [19] or set according to the original paper if available. Please refer to Appendix A.5 for more details.

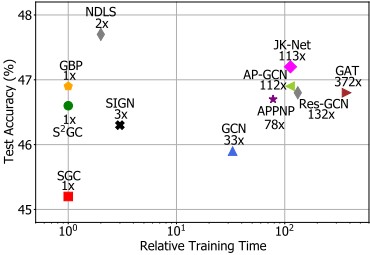

Figure 3: Performance along with training time on the Industry dataset.

Table 4: Results of inductive settings.

| Models | Flickr | Reddit |
|--------|--------|--------|
| GraphSAGE | 50.1±1.3 | 95.4±0.0 |
| FastGCN | 50.4±0.1 | 93.7±0.0 |
| ClusterGCN | 48.1±0.5 | 95.7±0.0 |
| GraphSAINT | 51.1±0.1 | 96.6±0.1 |
| NDLS-F+MLP* | 51.9±0.2 | 96.6±0.1 |
| GraphSAGE+NDLS-L* | 51.5±0.4 | 96.3±0.0 |
| NDLS* | **52.6±0.4** | **96.8±0.1** |

## 6.2 Experimental Results.

**End-to-end comparison.** To answer **Q1**, Table 3 and 4 show the test accuracy of considered methods in transductive and inductive settings. In the inductive setting, NDLS outperforms one of the most competitive baselines – GraphSAINT by a margin of 1.5% and 0.2% on Flickr and Reddit. NDLS exceeds the best GNN model among all considered baselines on each dataset by a margin of 0.2% to 0.8% in the transductive setting. In addition, we observe that with NDLS-L, the model performance of MLP, SGC, NDLS-F+MLP, and GraphSAGE can be further improved by a large margin. For example, the accuracy gain for MLP is 21.8%, 11.3%, 8.4%, and 5.6% on Cora, Citseer, PubMed, and Industry, respectively. To answer **Q2**, we replace the gate mechanism in the vanilla DAGNN with NDLS-L and refer to this method as DAGNN (NDLS-L). Surprisingly, DAGNN (NDLS-L) achieves at least comparable or (often) higher test accuracy compared with AP-GCN and DAGNN (Gate), and it shows that NDLS-L performs better than the learned mechanism in label smoothing. Furthermore, by replacing the original graph kernels with NDLS-F, NDLS-F+MLP outperforms both $S^2$GC and GBP on all compared datasets. This demonstrates the effectiveness of the proposed NDLS.

**Training Efficiency.** To answer **Q3**, we evaluate the efficiency of each method on a real-world industry graph dataset. Here, we pre-compute the smoothed features of each linear-model-based

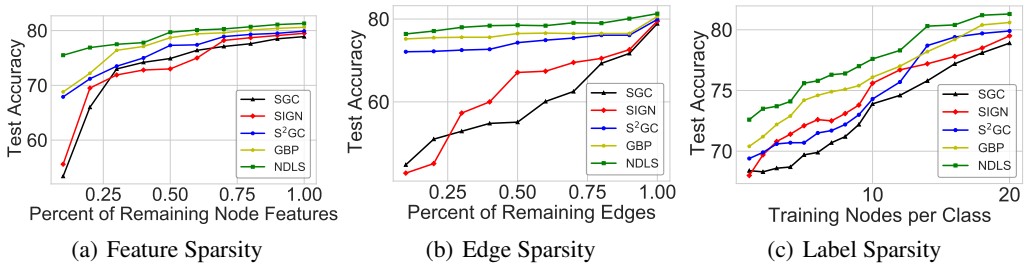

(a) Feature Sparsity      (b) Edge Sparsity      (c) Label Sparsity

Figure 4: Test accuracy on PubMed dataset under different levels of feature, edge and label sparsity.

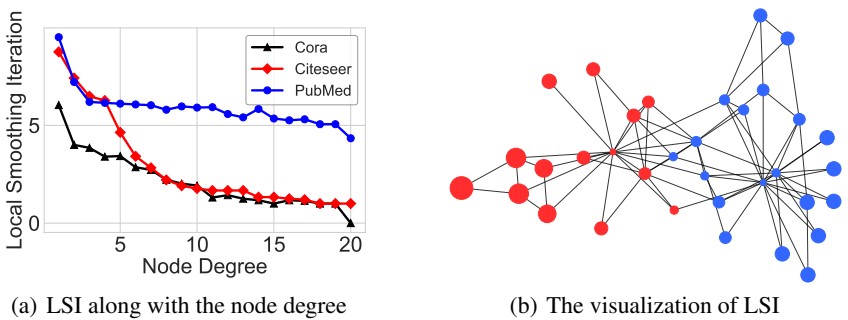

(a) LSI along with the node degree      (b) The visualization of LSI

Figure 5: (Left) LSI distribution along with the node degree in three citation networks. (Right) The visualization of LSI in Zachary's karate club network. Nodes with larger radius have larger LSIs.

GNN, and the time for pre-processing is also included in the training time. Figure 3 illustrates the results on the industry dataset across training time. Compared with linear-model-based GNNs, we observe that (1) both the coupled and decoupled GNNs require a significantly larger training time; (2) NDLS achieves the best test accuracy while consuming comparable training time with SGC.

**Performance on Sparse Graphs.** To reply **Q4**, we conduct experiments to test the performance of NDLS on feature, edge, and label sparsity problems. For feature sparsity, we assume that the features of unlabeled nodes are partially missing. In this scenario, it is necessary to calculate a personalized propagation iteration to "recover" each node's feature representation. To simulate edge sparsity settings, we randomly remove a fixed percentage of edges from the original graph. Besides, we enumerate the number of nodes per class from 1 to 20 in the training set to measure the effectiveness of NDLS given different levels of label sparsity. The results in Figure 4 show that NDLS outperforms all considered baselines by a large margin across different levels of feature, edge, and label sparsity, thus demonstrating that our method is more robust to the graph sparsity problem than the linear-model-based GNNs.

**Interpretability.** As mentioned by **Q1**, we here answer why NDLS is effective. One theoretical property of LSI is that the value correlates with the node degree negatively. We divide nodes into several groups, and each group consists of nodes with the same degree. And then we calculate the average LSI value for each group in the three citation networks respectively. Figure 5(a) depicts that nodes with a higher degree have a smaller LSI, which is consistent with Theorem 3.1. We also use NetworkX [10] to visualize the LSI in Zachary's karate club network [31]. Figure 5(b), where the radius of each node corresponds to the value of LSI, shows three interesting observations: (1) nodes with a larger degree have smaller LSIs; (2) nodes in the neighbor area have similar LSIs; (3) nodes adjacent to a super-node have smaller LSIs. The first observation is consistent with Theorem 3.1, and the latter two observations show consistency with Theorem 3.2.

# 7 Conclusion

In this paper, we present node-dependent local smoothing (NDLS), a simple and scalable graph learning method based on the local smoothing of features and labels. NDLS theoretically analyzes

what influences the smoothness and gives a bound to guide how to control the extent of smoothness for different nodes. By setting a node-specific smoothing iteration, each node in NDLS can smooth its feature/label to a local-smoothing state and then help to boost the model performance. Extensive experiments on seven real-world graph datasets demonstrate the high accuracy, scalability, efficiency, and flexibility of NDLS against the state-of-the-art GNNs.

## Broader Impact

NDLS can be employed in areas where graph modeling is the foremost choice, such as citation networks, social networks, chemical compounds, transaction graphs, road networks, etc. The effectiveness of NDLS when improving the predictive performance in those areas may bring a broad range of societal benefits. For example, accurately predicting the malicious accounts on transaction networks can help identify criminal behaviors such as stealing money and money laundering. Prediction on road networks can help avoid traffic overload and save people's time. A significant benefit of NDLS is that it offers a node-dependent solution. However, NDLS faces the risk of information leakage in the smoothed features or labels. In this regard, we encourage researchers to understand the privacy concerns of NDLS and investigate how to mitigate the possible information leakage.

## Acknowledgments and Disclosure of Funding

This work is supported by NSFC (No. 61832001, 61972004), Beijing Academy of Artificial Intelligence (BAAI), PKU-Baidu Fund 2019BD006, and PKU-Tencent Joint Research Lab. Zhi Yang and Bin Cui are the corresponding authors.

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
