# Node Dependent Local Smoothing for Scalable Graph Learning

**Wentao Zhang**[1], **Mingyu Yang**[1], **Zeang Sheng**[1], **Yang Li**[1]
**Wen Ouyang**[2], **Yangyu Tao**[2], **Zhi Yang**[1,3], **Bin Cui**[1,3,4]
[1]School of CS, Peking University  [2]Tencent Inc.
[3] Key Lab of High Confidence Software Technologies, Peking University
[4]Institute of Computational Social Science, Peking University (Qingdao), China
[1]{wentao.zhang, ymyu, shengzeang18, liyang.cs, yangzhi, bin.cui}@pku.edu.cn
[2]{gdpouyang, brucetao}@tencent.com

## A   Appendix

### A.1   Proofs of Theorems

We represent the adjacency matrix and the diagonal degree matrix of graph $\mathcal{G}$ by $A$ and $D$ respectively, represent $D+I$ and $A+I$ by $\tilde{D}$ and $\tilde{A}$. Then we denote $\tilde{D}^{-1}\tilde{A}$ as a transition matrix $P$. Suppose $P$ is connected, which means the graph is connected, for any initial distribution $\pi_0$, let

$$\tilde{\pi}(\pi_0) = \lim_{k \to \infty} \pi_0 P^k, \tag{1}$$

then according to [7], for any initial distribution $\pi_0$

$$\tilde{\pi}(\pi_0)_i = \frac{1}{n} \sum_{j=1}^{n} P_{ji}, \tag{2}$$

where $\tilde{\pi}_i$ denotes the $i^{th}$ component of $\tilde{\pi}(\pi_0)$, and $n$ denotes the number of nodes in graph. If matrix $P$ is unconnected, we can divide $P$ into connected blocks. Then for each blocks(denoted as $B_g$), there always be

$$\tilde{\pi}(\pi_0)_i = \frac{1}{n_g} \sum_{j \in B_g} P_{ji} * \sum_{j \in B_g} \pi_{0j}, \tag{3}$$

where $n_g$ is the number of nodes in $B_g$. To make the proof concise, we will assume matrix $P$ is connected, otherwise we can perform the same operation inside each block. Therefore, $\tilde{\pi}$ is independent to $\pi_0$, thus we replace $\tilde{\pi}(\pi_0)$ by $\tilde{\pi}$.

**Definition A.1** (**Local Mixing Time**). *The local mixing time (parameterized by $\epsilon$) with an initial distribution is defined as*

$$T(\pi_0, \epsilon) = \min\{t : ||\tilde{\pi} - \pi_0 P^t||_2 < \epsilon\}, \tag{4}$$

*where "$|| \cdot ||_2$" symbols two-nor m.*

In order to consider the impact of each node to the others separately, let $\pi_0 = e_i$, where $e_i$ is a one-hot vector with the $i^{th}$ component equal to 1, and the other components equal to 0. According to [6] we have lemma A.1.

**Lemma A.1.**

$$|(e_i P^t)_j - \tilde{\pi}_j| \leq \sqrt{\frac{\tilde{d}_j}{\tilde{d}_i}} \lambda_2^t, \tag{5}$$

35th Conference on Neural Information Processing Systems (NeurIPS 2021).

where $\lambda_2$ is the second large eigenvalue of $P$ and $\tilde{d}_i$ denotes the degree of node $v_i$ plus 1 (to include itself).

$$\tilde{d}_i = d_i + 1, \quad \tilde{d}_j = d_j + 1,$$

**Theorem A.2.**

$$T(e_i, \epsilon) \leq \log_{\lambda_2}(\epsilon\sqrt{\frac{\tilde{d}_i}{2m+n}}), \tag{6}$$

where $m$ and $n$ denote the number of edges and nodes in graph $\mathcal{G}$ separately.

$$\tilde{d}_i = d_i + 1,$$

*Proof.* [6] shows that when $\pi_0 = e_i$,

$$|(e_i P^t)_j - \tilde{\pi}_j| \leq \sqrt{\frac{\tilde{d}_j}{\tilde{d}_i}}\lambda_2^t, \tag{7}$$

where $(e_i P^t)_j$ symbols the $j^{th}$ element of $e_i P^t$. We denote $e_i P^t$ as $\pi_i(t)$, then

$$||\tilde{\pi} - \pi_i(t)||_2^2 = \sum_{j=1}^{n}(\tilde{\pi}_j - \pi_i(t)_j)^2$$

$$\leq \frac{\sum_{j=1}^{n}\tilde{d}_j}{\tilde{d}_i}\lambda_2^{2t} = \frac{2m+n}{\tilde{d}_i}\lambda_2^{2t}, \tag{8}$$

which means

$$||\tilde{\pi} - \pi_i(t)||_2 \leq \sqrt{\frac{2m+n}{\tilde{d}_i}}\lambda_2^t. \tag{9}$$

Now let

$$\epsilon = \sqrt{\frac{2m+n}{\tilde{d}_i}}\lambda_2^t,$$

there exists

$$T(e_i, \epsilon) \leq \log_{\lambda_2}(\epsilon\sqrt{\frac{\tilde{d}_i}{2m+n}}).$$

$\square$

Next consider the real situation in SGC with $n \times m$-dimension matrix $X(0)$ as input, where $n$ is the number of nodes, $m$ is the number of features. We apply $P$ as the normalized adjacent matrix.(The definition of $P$ is the same as $\tilde{\mathbf{A}}$ in main text). In feature propagation we have

$$X(t) = P^t X(0),$$

Now consider the $h^{th}$ feature of $X$, we define an $n \times n$ influence matrix

$$I_{hij}(t) = \frac{\partial X(t)_{ih}}{\partial X(0)_{jh}}, \tag{10}$$

Because $I_h(k)$ is independent to $h$, we replace $I_h(k)$ by $I(k)$, which can be formulated as

$$I(k) = I_h(k), \quad \forall h \in \{1, 2, .., f\}, \tag{11}$$

where $f$ symbols the number of features of $X$.

**Definition A.2** (**Local Smoothing Iteration**). *The Local Smoothing Iteration (parameterized by $\epsilon$) is defined as*

$$K(i, \epsilon) = \min\{k : ||\tilde{I}_i - I_i(k)||_2 < \epsilon\}. \tag{12}$$

According to Theorem A.2, there exists

**Theorem A.3** (**Theorem 3.1 in main text**). *When the normalized adjacent matrix is P,*

$$K(i, \epsilon) \leq \log_{\lambda_2}(\epsilon\sqrt{\frac{\tilde{d}_i}{2m+n}}). \tag{13}$$

*Proof.* From equation (9) we can derive that

$$||e_iP^\infty - e_iP^k||_2 \leq \sqrt{\frac{2m+n}{\tilde{d}_i}}\lambda_2^k.$$

Because

$$I_i(k) = P_i^k = e_iP^k \quad I_i(\infty) = P_i^\infty = e_iP^\infty,$$

we have

$$||I_i(\infty) - I_i(k)||_2 \leq \sqrt{\frac{2m+n}{\tilde{d}_i}}\lambda_2^k.$$

Now let

$$\epsilon = \sqrt{\frac{2m+n}{\tilde{d}_i}}\lambda_2^k,$$

there exists

$$K(i, \epsilon) \leq \log_{\lambda_2}(\epsilon\sqrt{\frac{\tilde{d}_i}{2m+n}}).$$

$\square$

Therefore, we expand Theorem A.3 to the propagation in SGC or our method. What is remarkable, Theorem A.3 requires $P$, which is equal to $\tilde{D}^{-1}\tilde{A}$ as the normalized adjacent matrix.

From Theorem A.3 we can conclude that the node which has a lager degree may need more steps to propagate. At the same time, we have another bond of local mixing time as following.

**Theorem A.4.** *For each node $v_i$ in graph $\mathcal{G}$, there always exits*

$$T(e_i, \epsilon) \leq \max\{T(e_j, \epsilon), j \in N(i)\} + 1. \tag{14}$$

*where N(i) is the set of node $v_i$'s neighbours.*

*Proof.*

$$|||\tilde{\pi} - e_iP^{t+1}||_2 = \frac{1}{|N(i)|}\sum_{j \in N(i)} ||\tilde{\pi} - e_jP^t||_2$$
$$\leq \max_{j \in N(i)} ||\tilde{\pi} - e_jP^t||_2. \tag{15}$$

Therefore, when

$$\max_{j \in N(i)} ||\tilde{\pi} - e_jP^t||_2 \leq \epsilon,$$

there exists

$$||\tilde{\pi} - e_iP^{t+1}||_2 \leq \epsilon.$$

Thus we can derive that

$$T(e_i, \epsilon) \leq \max\{T(e_j, \epsilon), j \in N(i)\} + 1.$$

$\square$

As we extend Theorem A.2 to Theorem A.3, according to Theorem A.4, there always be

**Theorem A.5** (**Theorem 3.2 in main text**). *For each node $v_i$ in graph $\mathcal{G}$, there always exits*

$$K(i, \epsilon) \leq \max\{K(j, \epsilon), j \in N(i)\} + 1. \tag{16}$$

Table 1: Results of different base models on PubMed.

| Base Models | Models | Accuracy | Gain |
|---|---|---|---|
| MLP | Base | 72.7±0.6 | - |
| | + NDLS-F | 81.1±0.6 | + 8.4 |
| | + NDLS-L | 81.1±0.6 | + 8.4 |
| | + NDLS (both) | **81.4±0.4** | + 8.7 |
| RF | Base | 74.4±0.2 | - |
| | + NDLS-F | 80.3±0.1 | + 5.9 |
| | + NDLS-L | 80.0±0.2 | + 5.6 |
| | + NDLS (both) | **80.5±0.4** | + 6.1 |
| XGB | Base | 74.1±0.2 | - |
| | + NDLS-F | 81.0±0.3 | + 6.9 |
| | + NDLS-L | 79.8±0.2 | + 5.7 |
| | + NDLS (both) | **81.6±0.3** | + 7.5 |

## A.2 Results with More Base Models

Our proposed NDLS consists of three stages: (1) feature smoothing with NDLS (NDLS-F), (2) model training with smoothed features, and (3) label smoothing with NDLS (NDLS-L). In stage (2), the default option of the base model is a Multilayer Perceptron (MLP). Besides MLP, many other models can also be used in stage (2) to generate soft labels. To verify it, here we replace the MLP in stage (2) with popular machine learning models Random Forest [11] and XGBoost [4], and measure their node classification performance on PubMed dataset. The experiment results are shown in Table 1 where Random Forest and XGBoost are abbreviated as *RF* and *XGB* respectively.

Compared to the vanilla model, both Random Forest and XGBoost achieve significant performance gain with the addition of our NDLS. With the help of NDLS, Random Forest and XGBoost outperforms their base models by $6.1\%$ and $7.5\%$ respectively. From Table 1, we can observe that both NDLS-F and NDLS-L can contribute great performance boost to the base model, where the gains are at least $5\%$. When all equipped with both NDLS-F and NDLS-L, XGBoost beat the default MLP, achieving a test accuracy of $81.6\%$. Although Random Forest – $80.5\%$ – cannot outperform the other two models, it is still a competitive model.

The above experiment demonstrates that the base model selection in stage (2) is rather flexible in our NDLS. Both traditional machine learning methods and neural networks are promising candidates in the proposed method.

## A.3 Dataset Description

**Cora**, **Citeseer**, and **Pubmed**[1] are three popular citation network datasets, and we follow the public training/validation/test split in GCN [9]. In these three networks, papers from different topics are considered as nodes, and the edges are citations among the papers. The node attributes are binary word vectors, and class labels are the topics papers belong to.

**Reddit** is a social network dataset derived from the community structure of numerous Reddit posts. It is a well-known inductive training dataset, and the training/validation/test split in our experiment is the same as the one in GraphSAGE [8].

---

[1]https://github.com/tkipf/gcn/tree/master/gcn/data

Table 2: URLs of baseline codes.

| Type | Baselines | URLs |
|---|---|---|
| Coupled | GCN
GAT | https://github.com/rusty1s/pytorch_geometric
https://github.com/rusty1s/pytorch_geometric |
| Decoupled | APPNP
PPRGo
AP-GCN
DAGNN | https://github.com/rusty1s/pytorch_geometric
https://github.com/TUM-DAML/pprgo_pytorch
https://github.com/spindro/AP-GCN
https://github.com/divelab/DeeperGNN |
| Sampling | GraphSAGE
GraphSAINT
FastGCN
Cluster-GCN | https://github.com/williamleif/GraphSAGE
https://github.com/GraphSAINT/GraphSAINT
https://github.com/matenure/FastGCN
https://github.com/benedekrozemberczki/ClusterGCN |
| Linear | SGC
SIGN
S$^2$GC
GBP | https://github.com/Tiiiger/SGC
https://github.com/twitter-research/sign
https://github.com/allenhaozhu/SSGC
https://github.com/chennnM/GBP |

**Flickr** originates from NUS-wide [2] and contains different types of images based on the descriptions and common properties of online images. The public version of Reddit and Flickr provided by GraphSAINT[3] is used in our paper.

**Industry** is a short-form video graph, collected from a real-world mobile application from our industrial cooperative enterprise. We sampled 1,000,000 users and videos from the app, and treat these items as nodes. The edges in the generated bipartite graph represent that the user clicks the short-form videos. Each user has 64 features and the target is to category these short-form videos into 253 different classes.

**ogbn-papers100M** is a directed citation graph of 111 million papers indexed by MAG [16]. Among its node set, approximately 1.5 million of them are arXiv papers, each of which is manually labeled with one of arXiv's subject areas. Currently, this dataset is much larger than any existing public node classification datasets.

### A.4 Compared Baselines

The main characteristic of all baselines are listed below:

- **GCN** [9]: GCN is a novel and efficient method for semi-supervised classification on graph-structured data.
- **GAT** [15]: GAT leverages masked self-attention layers to specify different weights to different nodes in a neighborhood, thus better represent graph information.
- **JK-Net** [18]: JK-Net is a flexible network embedding method that could gather different neighborhood ranges to enable better structure-aware representation.
- **APPNP** [10]: APPNP uses the relationship between graph convolution networks (GCN) and PageRank to derive improved node representations.
- **AP-GCN** [14]: AP-GCN uses a halting unit to decide a receptive range of a given node.
- **DAGNN** [12]: DAGNN proposes to decouple the representation transformation and propagation, and show that deep graph neural networks without this entanglement can leverage large receptive fields without suffering from performance deterioration.
- **PPRGo** [1]: utilizes an efficient approximation of information diffusion in GNNs resulting in significant speed gains while maintaining state-of-the-art prediction performance.

---

[2]http://lms.comp.nus.edu.sg/research/NUS-WIDE.html
[3]https://github.com/GraphSAINT/GraphSAINT

Table 3: Performance comparison between C&S and NDLS-L

| Methods | Cora | Citeseer | PubMed | ogbn-papers100M |
|---|---|---|---|---|
| MLP+C&S | 87.2 | 76.6 | 88.3 | 63.9 |
| MLP+NDLS-L | **88.1** | **78.3** | **88.5** | **64.6** |

- **GraphSAGE** [8]: GraphSAGE is an inductive framework that leverages node attribute information to efficiently generate representations on previously unseen data.

- **FastGCN** [2]: FastGCN interprets graph convolutions as integral transforms of embedding functions under probability measures.

- **Cluster-GCN** [5]: Cluster-GCN is a novel GCN algorithm that is suitable for SGD-based training by exploiting the graph clustering structure.

- **GraphSAINT** [19]: GraphSAINT constructs mini-batches by sampling the training graph, rather than the nodes or edges across GCN layers.

- **SGC** [17]: SGC simplifies GCN by removing nonlinearities and collapsing weight matrices between consecutive layers.

- **SIGN** [13]: SIGN is an efficient and scalable graph embedding method that sidesteps graph sampling in GCN and uses different local graph operators to support different tasks.

- **S$^2$GC** [20]: S$^2$GC uses a modified Markov Diffusion Kernel to derive a variant of GCN, and it can be used as a trade-off of low-pass and high-pass filter which captures the global and local contexts of each node.

- **GBP** [3]: GBP utilizes a localized bidirectional propagation process from both the feature vectors and the training/testing nodes

Table 2 summarizes the github URLs of the compared baselines. Following the original paper, we implement JK-Net by ourself since there is no official version available.

### A.5 Implementation Details

**Hyperparameter details.** In stage (1), when computing the Local Smoothing Iteration, the maximal value of $k$ in equation (12) is set to 200 and the optimal $\epsilon$ value is get by means of a grid search from {0.01, 0.03, 0.05}. In stage (2), we use a simple two-layer MLP to get the base prediction. The hidden size is set to 64 in small datasets – Cora, Citeseer and Pubmed. While in larger datasets – Flicker, Reddit, Industry and ogbn-papers100M, the hidden size is set to 256. As for the dropout percentage and the learning rate, we use a grid search from {0.2, 0.4, 0.6, 0.8} and {0.1, 0.01, 0.001} respectively. In stage (3), during the computation of the Local Smoothing Iteration, the maximal value of $k$ is set to 40. The optimal value of $\epsilon$ is obtained through the same process in stage (1).

**Implementation environment.** The experiments are conducted on a machine with Intel(R) Xeon(R) Gold 5120 CPU @ 2.20GHz, and a single NVIDIA TITAN RTX GPU with 24GB memory. The operating system of the machine is Ubuntu 16.04. As for software versions, we use Python 3.6, Pytorch 1.7.1 and CUDA 10.1.

### A.6 Comparison and Combination with Correct&Smooth

Similar to our NDLS-L, Correct and Smooth (C&S) also applies post-processing on the model prediction. Therefore, we compare NDLS-L with C&S below.

**Adaptivity to node.** C&S adopts a propagation scheme based on Personalized PageRank (PPR), which always maintains certain input information to slow down the occurrence of over-smoothing. The expected number of smoothing iterations is controlled by the restart probability, which is a constant for all nodes. Therefore, C&S still falls into the routine of fixed smoothing iteration. Instead, NDLS-L employs node-specific smoothing iterations. We compare each method's performance (test accuracy, %) under the same data split as in the C&S paper (60%/20%/20% on three citation

Table 4: Performance comparison under varied label rate on the Cora dataset.

| Methods | 2% | 5% | 10% | 20% | 40% | 60% |
|---|---|---|---|---|---|---|
| MLP+S | 63.1 | 77.8 | 82.6 | 84.2 | 85.4 | 86.4 |
| MLP+C&S | 62.8 | 76.7 | 82.8 | 84.9 | 86.4 | 87.2 |
| MLP+NDLS-L | **77.4** | **83.9** | **85.3** | **86.5** | **87.6** | **88.1** |

Table 5: Performance comparison after combining the node-dependent idea with C&S.

| Methods | Cora | Citeseer | PubMed |
|---|---|---|---|
| MLP+C&S | 76.7 | 70.8 | 76.5 |
| MLP+C&S+nd | **79.9** | **71.1** | **78.4** |

Table 6: Efficiency comparison on the PubMed dataset.

| | SGC | S$^2$GC | GBP | NDLS | SIGN | JK-Net | DAGNN | GCN | ResGCN | APPNP | GAT |
|---|---|---|---|---|---|---|---|---|---|---|---|
| Time | 1.00 | 1.19 | 1.20 | 1.50 | 1.59 | 11.42 | 14.39 | 20.43 | 20.49 | 28.88 | 33.23 |
| Accuracy | 78.9 | 79.9 | 80.6 | **81.4** | 79.5 | 78.8 | 80.5 | 79.3 | 78.6 | 80.1 | 79.0 |

networks, official split on ogbn-papers100M), and the experimental results in Table 3 show that NDLS-L outperforms C&S in different datasets.

**Sensitivity to label rate.** During the "Correct" stage, C&S propagates uncertainties from the training data across the graph to correct the base predictions. However, the uncertainties might not be accurate when the number of training nodes is relatively small, thus even degrading the performance. To confirm the above assumption, we conduct experiments on the Cora dataset under different label rates, and the experimental results are provided in Table 4. As illustrated, the result of C&S drops much faster than NDLS-L's when the label rate decreases. What's more, MLP+S (removing the "Correct" stage) outperforms MLP+C&S when the label rate is low as expected.

Compared with C&S, NDLS is more general in terms of smoothing types. C&S can only smooth label predictions. Instead, NDLS can smooth both node features and label predictions and combine them to boost the model performance further.

**Node Adaptive C&S.** The node-dependent mechanism in our NDLS can easily be combined with C&S. The two stages of C&S both contain a smoothing process using the personalized PageRank matrix, where a coefficient controls the remaining percentage of the original node feature. Here, we can precompute the smoothed node features after the same smoothing step yet under different values like 0.1, 0.2, ..., 0.9. After that, we adopt the same strategy in our NDLS: for each node, we choose the first in the ascending order that the distance from the smoothed node feature to the stationarity is less than a tuned hyperparameter. By this means, the smoothing process in C&S can be carried out in a node-dependent way.

We also evaluate the performance of C&S combined with the node-dependent idea (represented as C&S+nd) on the three citation networks under official splits, and the experimental results in Table 5 show that C&S combined with NDLS consistently outperforms the original version of C&S.

## A.7 Training Efficiency Study

we measure the training efficiency of the compared baselines on the widely used PubMed dataset. Using the training time of SGC as the baseline, the relative training time and the corresponding test accuracy of NDLS and the baseline methods are shown in Table 6. Compared with other baselines, NDLS can get the highest test accuracy while maintaining competitive training efficiency.

Table 7: Performance comparison on the ogbn-arxiv dataset.

| | MLP | MLP+C&S | GCN | SGC | SIGN | DAGNN | JK-Net | S$^2$GC | GBP | NDLS | GAT |
|---|---|---|---|---|---|---|---|---|---|---|---|
| Accuracy | 55.50 | 71.58 | 71.74 | 71.72 | 71.95 | 72.09 | 72.19 | 72.21 | 72.45 | 73.04 | **73.56** |

## A.8 Experiments on ogbn-arxiv

We also conduct experiments on the ogbn-arxiv dataset. The experiment results (test accuracy, %) are provided in Table 7. Although GAT outperforms NDLS on ogbn-arxiv dataset, it is hard to scale to large graphs like ogbn-papers100M dataset. Note that MLP+C&S on the OGB leaderboard makes use of not only the original node feature but also diffusion embeddings and spectral embeddings. Here we remove the latter two embeddings for fairness, and the authentic MLP+C&S achieves 71.58% on the ogbn-arxiv dataset.