# OpenReview forum: "Node Dependent Local Smoothing for Scalable Graph Learning"
_NeurIPS.cc/2021/Conference — NeurIPS 2021 Spotlight_

### Official Review · Reviewer_yciW · 2021-07-12

**Rating:** 7
**Confidence:** 4

**Summary:**

This work proposes NDLS, a strategy to estimate a different number of 'propagations' for each node in the graph, delimiting the neighborhoods used to update node embeddings. The authors apply NDLS both as a pre-processing (feature space) and post-processing (prediction space) technique. Between pre-processing and post-processing, any vanilla (non-Graph) classifier can be used.

**Limitations And Societal Impact:**

The authors adequately discussed limitations.

**Main Review:**

Intuitively, NDLS chooses neighborhood sizes so that all nodes are equally far from oversmoothing. Overall, I like this idea and its simplicity. Scalability is also an asset, and the idea for computing node-dependent numbers of propagations is intuitive. I think this work fits well in the current trend of simplifying graph ML. The idea is significantly novel and the technical developments appear sound. The writing is passable.

### Questions and issues
* How does this work compare to https://arxiv.org/abs/2010.13993 ?

* Why are the results in Table 3 different from the ones in the OGB leaderboard? For instance, Table 3 shows SIGN achieving $64.2$ average accuracy on ogbn-papers100M. However, OGB leaderboards (https://ogb.stanford.edu/docs/leader_nodeprop/) show $65.68$.

* Previous works (https://arxiv.org/abs/2003.00982) have shown that small datasets --- such as Cora, Citeseer and Pubmed --- are not reliable benchmarks. Do you have results for other OGB large-scale benchmarks?

* Are the bounds in section 3 used only for interpretation or to avoid directly computing $k$?

* How is $k(\cdot, \cdot)$ computed in practice and how expensive is it? Does the method require computation of eigenvalues (as in Theorem 3.1, eq 5)?

* As a curiosity, how important is the normalization factor in Eq. 8? Do you have any ablation studies for this?

* In the sparse graph experiments, what do the authors do with missing features? Fill with some default value?

* Can the authors add more plots like Figure 3 (times vs. performance) either in the manuscript or appendix?


### Typos, etc
* The authors often use acronyms without introducing them (e.g. in the abstract and first paragraph of intro)
* Line 40: 'poorly connected nodes needs' -> poorly connected nodes need
* Lines 56-57: authors say real-world graphs have long-tailed degree distributions. Not all real-world graphs do have it. Probably better to be careful.
* Line 75 and 76: a matrix should not be defined as a set


**Time Spent Reviewing:**

8

---

> ### Author Response · Authors · 2021-08-10
> **Response to Reviewer-4 yciW**
>
> Thanks for your review and feedback. We sincerely appreciate your assessment of our work as "fits well in the current trend" and "significantly novel". As suggested by the reviewer, we provide the following responses.
>
>
> ### 1. Comparison to C&S:
> The reviewer-1 1k7Z also proposes the same question about the comparison with the Correct and Smooth (C&S) method. Compared to C&S, NDLS exhibits three major advantages: **node-dependent aggregation**, **less sensitive to low label rate**, and **more general smoothing operations**. We also conduct experiments to show that NDLS outperforms C&S. Please refer to our response to reviewer-1 1k7Z for the details.
>
>
> ### 2. SIGN result:
> On the OGB leaderboard, SIGN obtains the test accuracy of 65.68% by applying and integrating multiple filter matrices to smooth the node features, including *standard normalized adjacency matrix*, *PPR diffusion matrix*, and *normalized triangle-induced adjacency matrix*. For a **fair comparison** in **Table 3**, we use SIGN with **the normalized adjacency matrix only** like NDLS and all the other baselines. It obtains the test accuracy of 64.2%, which *is consistent with SIGN's original paper* (the reported accuracy using the normalized adjacency matrix only is 64.28%).
>
>
> ### 3. Other OGB benchmarks:
> Thanks for your suggestion. We also conduct experiments on the ogbn-arxiv dataset. The experiment results (test accuracy, %) are provided in the table below. Although GAT outperforms NDLS on ogbn-arxiv dataset, it is **hard to scale to large graphs** like ogbn-papers100M dataset. Note that MLP+C&S on the OGB leaderboard makes use of not only the original node feature but also diffusion embeddings and spectral embeddings. Here we *remove the latter two embeddings for fairness*, and the authentic MLP+C&S achieves 71.58% on the ogbn-arxiv dataset.
>
> |            | MLP   | MLP+C&S |  GCN  |  SGC  | SIGN  | DAGNN | JK-Net | S$^2$GC |  GBP  | NDLS  |  GAT  |
> |:----------| ----- |:-------:|:-----:|:-----:|:-----:|:-----:|:------:|:-------:|:-----:|:-----:|:-----:|
> | ogbn-arxiv | 55.50 |  71.58  | 71.74 | 71.72 | 71.95 | 72.09 | 72.19  |  72.21  | 72.45 | **73.04** | 73.56 |
>
>
> ### 4. LSI Bounds:
> NDLS **does not directly use** the upper bound in the theorem to compute the exact LSI. The bound in our theorem is proposed to provide an in-depth understanding of **which factors affect LSI**.
>
>
> ### 5. The computation of $\small K(i, \epsilon)$:
> NDLS calculates $\small K(i, \epsilon)$ based on **Eq.4**, where over-smoothing stationarity can be obtained by **Eq.2**, and step-wise influence matrix $\small I(k)$ can be obtained by $\small \hat{A}^k$. Here, $\small \hat{A}$ is the normalized adjacency matrix.
>
> However, for very large-scale graphs such as ogbn-papers100M, we adopt an approximation approach that integrates a dimension reduction procedure by replacing $\small ||\tilde{I}_i-I(k)_i||_2$ in **Eq.4** with $\small ||A^{k}X-A^{\infty}X||_2$. Here, $\small X$ is the original node feature matrix of the dataset.
>
> In this way, the high $\small O(M\*N)$ time complexity of computing $\small A\*A^{k-1}$ for $\small k < K$ can be reduced to $\small O(M\*d)$ of computing $\small A\*(A^{k-1}X)$ for $\small k < K$, where $\small K$ is the maximal iteration number. $\small N$ is the number of nodes, $\small M$ is the number of edges, and $d$ is the dimensionality of the node feature (which is usually small compared to $\small N$). After the matrix multiplications, the cost for the remaining computation to find LSI is  $\small O(K\*N\*N)$ on small graphs and $\small O(K\*N\*d)$ on large graphs.
>
>
> ### 6. Normalization factor:
> Our original intention of performing averaging (normalization in your question) is to keep each node's generated features (i.e., final embeddings) on the **same scale**. Without averaging, a node with LSI=9 might have higher feature values than a node with LSI=4, which could affect the model's performance. As per your suggestion, we perform an ablation study for this. The results (test accuracy, %) are provided in the table below, demonstrating such normalization's effectiveness.
>
> |   Methods     | Cora | Citeseer | PubMed |
> |:----------------------|:----:|:--------:|:------:|
> | NDLS without normalization | 81.6 | 71.3 | 80.5 |
> | NDLS with normalization  | **84.6** | **73.7** | **81.4** |
>
>
> ### 7. Missing features:
> We **fill the missing features with a default value of zero**, similar to the following previous works:
>
> [Zhao et al. 2020] Lingxiao Zhao, Leman Akoglu. [*PairNorm: Tackling Oversmoothing in GNNs.*](https://openreview.net/forum?id=rkecl1rtwB) ICLR 2020.
>
> [Zhou et al. 2020] Kaixiong Zhou, Xiao Huang, Yuening Li, Daochen Zha, Rui Chen, Xia Hu. [*Towards Deeper Graph Neural Networks with Differentiable Group Normalization.*](https://proceedings.neurips.cc/paper/2020/hash/33dd6dba1d56e826aac1cbf23cdcca87-Abstract.html) NeurIPS 2020.
>
>
> ### 8. More plots:
> Thanks for your suggestion. We also conduct experiments on the widely used dataset PubMed to add more empirical results like **Figure 3**, and we will update this in our revised manuscript. Using the training time of SGC as the baseline, the relative training time and the test accuracy of NDLS and the baseline methods are provided in the table below.
>
> |               | SGC  | S$^2$GC | GBP  | NDLS | SIGN | JK-Net | DAGNN |  GCN  | ResGCN | APPNP |  GAT  |
> |:-------------|:----:|:-------:|:----:|:----:|:----:|:------:|:-----:|:-----:|:------:|:-----:|:-----:|
> | Relative Time | 1.00 |  1.19   | 1.20 | **1.50** | 1.59 | 11.42  | 14.39 | 20.43 | 20.49  | 28.88 | 33.23 |
> | Test Accuracy (%) | 78.9 |  79.9   | 80.6 | **81.4** | 79.5 |  78.8  | 80.5  | 79.3  |  78.6  | 80.1  | 79.0  |
>
>
> ### 9. Typos:
> Thank you for reminding us of the typos. We will correct them in the final manuscript.

---

> ### Author Response · Authors · 2021-09-02
> **Response to Reviewer-4 yciW**
>
> We really appreciate your helpful and valuable reviews, especially the advice of 1) comparison with C&S 2) experiments on other OGB large-scale benchmarks 3) more plots like Figure 3.
>
> We have carefully responded to your questions and added the experiments as suggested.
>
> We hope to address all your points, and we are happy to respond if you have additional comments or concerns on our responses.
>
> Respectfully,
>
> Paper1237 Authors

---

> > ### Comment · Reviewer_yciW · 2021-09-03
> > **Raising score**
> >
> > Thank you for the answers, which appropriately address my concerns. I also appreciate the novel experiments in an OGB dataset, the comparison with C&S, the ablation with a different aggregation, and the time comparisons for a novel dataset. I increased the score to 7.

---

### Official Review · Reviewer_2avU · 2021-07-15

**Rating:** 6
**Confidence:** 3

**Summary:**

In this paper, the authors proposed a GNN structure name NDLS which comprises three parts: i) the feature propagation part, NDLS-F, ii) a MLP for label prediction and iii) a post processing label smoothing part NDLS-L. The first and last part and are done individually as pre-processing and post-processing, where each node propagates its feature/label representation according to their local smooth iteration (LSI). The LSI is computed per node based on its connectivity and designed to avoid over-smoothing instead of using same step for all nodes. Due to the simplified framework, the model can be trained efficiently without losing performance when compared to SOTA baselines.

**Limitations And Societal Impact:**

The authors did not addressed the possible limitation and negative societal impact of their work. Protecting personal privacy especially in social network data is always worth discussion.

**Main Review:**

The paper is very well-written and easy to follow, with intuitive figures for better understanding the concept. The authors did extensive experiments and the results are nicely interpreted.

pros:
- The difference of LSI distribution in different datasets is a nice indicator of how over-smoothing is also a data-dependent issue and should be treated adaptively.
- Detailed experiments with analysis on how different component of the model affect the results.
- Efficient algorithm with separated component that can be precomputed.
- Intuitive interpretation of results and trained models.

cons:
- The definition of LSI in equation (4) implicitly assumes a monotonous decreasing distance with k, which I am not sure if this is the case.
- Improvement on inductive settings are marginal, and wether separate pre/post processing is needed is not clear.

**Time Spent Reviewing:**

6

---

> ### Author Response · Authors · 2021-08-10
> **Response to Reviewer-3 2avU**
>
> Thanks for your insightful feedback. We appreciate your assessment about this paper being "well written and easy to follow". The answers to your concerns are as follows.
>
>
> ### 1. Monotonous decreasing distance:
> The reviewer raises a good question. The definition of LSI does not require a decreasing distance with k, and we will try to explain why the LSI still works even if k is not strictly decreasing. We find that the distance is approximately (though not strictly) monotonous decreasing with k for most nodes. Concretely, **more than 95% of nodes** exhibit a monotonous decreasing distance with k across all datasets we used. It is because that a node tends to approach the stationarity as the smoothing iteration k grows, *except for a few nodes with large degrees*. For example, on the Cora dataset, over 97% of nodes exhibit a monotonous decreasing distance with k; only **73 out of 2708** nodes are out of this scope. Besides, the average node degree of these 73 nodes is 10.37, significantly larger than the node degree of the whole graph, 2.22. Further, we find these 73 nodes include all the nodes in the graph whose degrees are greater than 11. More formally, the proof of **Theorem A.3** in the appendix also shows that the upper bound of the distance is strictly monotonous decreasing with k.
>
>
> ### 2. Inductive settings:
> The improvement on the Reddit dataset seems marginal because *the current classification performance has already been extremely high* (i.e., higher than 95%) on this dataset. Our proposed NDLS **achieves the SOTA performance on the Reddit dataset**. Moreover, for more challenging inductive datasets such as Flickr, NDLS demonstrates a significant improvement in final accuracy.
>
>
> ### 3. Why separating pre/post-processing:
> We separate pre/post-processing due to the following reasons:
> 1. The stage decoupling of pre/post-processing and model training is widely used in scalable GNNs. For example, the pre-processing in SGC [Wu et al. 2019] and GBP [Chen et al. 2020] helps the models take orders of magnitude **less time to converge** and can easily **scale to large graphs**. **Figure 3** demonstrates the efficiency benefits of stage decoupling compared to those trained end-to-end (such as GCN [Kipf et al. 2017] and APPNP [Klicpera et al. 2018]).
> 2. As shown in **Table 4**, the experiment results illustrate that both pre/post-processing *can individually improve* the classification performance. The performance *can be further improved if both are applied simultaneously*.
>
> [Wu et al. 2019] Felix Wu, Tianyi Zhang, Amauri Holanda de Souza Jr., Christopher Fifty, Tao Yu, and Kilian Q. Weinberger. [*Simplifying graph convolutional networks.*](http://proceedings.mlr.press/v97/wu19e/wu19e.pdf) ICML 2019.
>
> [Chen et al. 2020] Ming Chen, Zhewei Wei, Bolin Ding, Yaliang Li, Ye Yuan, Xiaoyong Du, Ji-Rong Wen. [*Scalable Graph Neural Networks via Bidirectional Propagation.*](https://proceedings.neurips.cc/paper/2020/hash/a7789ef88d599b8df86bbee632b2994d-Abstract.html) NeurIPS 2020.
>
> [Huang et al. 2020] Qian Huang, Horace He, Abhay Singh, Ser-Nam Lim, and Austin R. Benson. [*Combining Label Propagation and Simple Models out-performs Graph Neural Networks.*](https://openreview.net/pdf?id=8E1-f3VhX1o) ICLR 2020.
>
> [Kipf et al. 2017] Thomas N. Kipf, and Max Welling. [*Semi-Supervised Classification with Graph Convolutional Networks.*](https://openreview.net/pdf?id=SJU4ayYgl) ICLR 2017.
>
> [Klicpera et al. 2018] Johannes Klicpera, Aleksandar Bojchevski, and Stephan Gunnemann. [*Predict then Propagate: Graph Neural Networks meet Personalized PageRank.*](https://openreview.net/pdf?id=H1gL-2A9Ym) ICLR 2018.
>
>
> ### 4. Societal impact:
> NDLS can be employed in areas where graph modeling is the foremost choice, such as citation networks, social networks, chemical compounds, transaction graphs, road networks, etc. The effectiveness of NDLS when improving the predictive performance in those areas may bring a broad range of societal benefits. For example, accurately predicting the malicious accounts on transaction networks can *help identify criminal behaviors such as stealing money and money laundering*. Prediction on road networks can *help avoid traffic overload and save people's time*. A significant benefit of NDLS is that it offers a node-dependent solution. However, NDLS faces the risk of **information leakage** in the smoothed features or labels. In this regard, we encourage researchers to understand the privacy concerns of NDLS and investigate how to mitigate the possible information leakage.

---

> ### Author Response · Authors · 2021-09-02
> **Response to Reviewer-3 2avU**
>
> Thanks for your helpful and insightful reviews.
>
> We have responded to "Monotonous decreasing distance", "Inductive settings" and "Why separating pre/post-processing" in our previous response. Besides, we have also added the possible limitation and negative societal impact of our work.
>
> We hope our response can solve the problems you proposed, and we are happy to respond if some problems still exist.
>
> Respectfully,
>
> Paper1237 Authors

---

### Official Review · Reviewer_8aSG · 2021-07-16

**Rating:** 6
**Confidence:** 4

**Summary:**

This paper proposed a node-dependent local smoothing method for graph neural networks via computing a local smoothing iteration value. Experimental evaluation was conducted on seven datasets.

**Limitations And Societal Impact:**

See the above.

**Main Review:**

The paper is well-written on the whole. Illustrations are intuitive and can help readers understand the proposed idea greatly. Nonetheless, there are several questions/problems as detailed below:

1. The use of acronyms, particularly in the abstract, is problematic to me. SGC and APPNP are acronyms unknown to me at least. Acronyms should be defined before using.

2. Notations are not always consistent. For example, I think the X in line 109 and 111 should be in bold normal font, as previously appeared.

3. Since the "Industry" dataset is not in the public domain, it will be helpful to provide an explanation why this dataset is chosen instead of another public dataset.

4. Could you do a paired t-test at least for GBP vs NDLS in Table 3 to study the statistical significance of the improvements?

5. Training efficiency study in Sec. 6.2 and Figure 3: Why to choose the private "Industry" dataset instead of a public dataset for this study. Do you have similar studies and plots for other datasets? Using a public dataset will make the result more reproducible.

6. From Figure 5(a), I am wondering whether using the node degree will be a simpler alternative than the proposed LSI for the proposed framework, e.g. 1/node_degree or similar. Have you studied such a simple choice?

7. Although Appendix A.5 provided hyperparameter details, it is not always clear how certain hyperparameters were chosen. For example, for k=200 in stage (1) and k=40 in stage (3), were these number just magically set by the authors or from tuning? If by tuning, how was it done? More importantly, I did not find studies on the performance sensitivity w.r.t. such hyperparameters.

8. Consistency issue: why is the caption for Table 4 at the bottom while captions for other tables are at the top?

9. Anonymity issue: the supplementary material is not fully anonymized. The readme shows a username with a local path "C:\Users\......\Desktop\NDLS", where I replaced the name with "......".



**Time Spent Reviewing:**

2

---

> ### Author Response · Authors · 2021-08-10
> **Response to Reviewer-2 8aSG**
>
> We appreciate your insightful review. Please read the following responses to your concerns for further improvement.
>
>
> ### 1. Acronyms & notation:
> Thanks for mentioning the acronym confusion and the notational inconsistency. We will correct them in the revised manuscript.
>
>
> ### 2. Why "Industry" dataset:
> Good question! The reason why we choose the Industry dataset instead of other public datasets is that the Industry dataset owns the following characteristics:
> 1. Unlike commonly-used citation graphs (Cora, Citeseer, PubMed, and ogbn-papers100M), the Industry dataset is **another type of graph** where nodes represent short-form videos and edges between two nodes represent the same user clicks these two videos.
> 2. Compared with other existing datasets, the Industry dataset is **closer to the real industrial setting**.
> 3. It provides a benchmark dataset that is **both large-scale and highly sparse**. To make the results reproducible and facilitate relevant research, *we will make this dataset publicly available along with the paper*.
>
>
> ### 3. Paired t-test:
> As suggested by the reviewer, we perform a paired t-test for GBP vs. NDLS in **Table 3**. The p-value is *less than 0.01*, demonstrating that the improvement is **statistically significant**.
>
>
> ### 4. Training efficiency study:
> We choose the Industry dataset for efficiency study because it **is closer to the real industrial setting**. To make the result more reproducible, *we will make this dataset publicly available*. As per your suggestion, we also conduct the same comparison on the widely used PubMed dataset, which shows a similar trend of efficiency and accuracy. Using the training time of SGC as the baseline, the relative training time and the test accuracy of NDLS and the baseline methods are as follows.
>
> |               | SGC  | S$^2$GC | GBP  | NDLS | SIGN | JK-Net | DAGNN |  GCN  | ResGCN | APPNP |  GAT  |
> |:-------------|:----:|:-------:|:----:|:----:|:----:|:------:|:-----:|:-----:|:------:|:-----:|:-----:|
> | Relative Time | 1.00 |  1.19   | 1.20 | **1.50** | 1.59 | 11.42  | 14.39 | 20.43 | 20.49  | 28.88 | 33.23 |
> | Test Accuracy (%) | 78.9 |  79.9   | 80.6 | **81.4** | 79.5 |  78.8  | 80.5  | 79.3  |  78.6  | 80.1  | 79.0  |
>
>
> ### 5. Can node degree be an alternative?
> The reviewer raises a good point. From **Theorem 3.1**, we see that the upper bound of LSI of a specific node is correlated negatively with its degree. However, the degree itself may be an ineffective alternative because **Eq.(7)** indicates that the exact **LSI is also affected by** 1) graph structure represented by *the second largest eigenvalue* and 2) LSI characteristics within *the local neighborhood*. Taking **Figure 5(b)** as an example, although some nodes have small degrees, their LSIs are also quite small because they are connected to supernodes.
>
> To explore whether **only using node degrees** is an acceptable alternative or not, we also conduct experiments and replace the LSI with *2\*max_degree/node_degree*. Performance results (test accuracy, %) of this alternative on Cora, Citeseer, and PubMed datasets are provided in the table below, which exceeds the performance of some baselines. However, this method still *falls behind the performance of NDLS* by a large margin and *lacks the theoretical basis* that NDLS has.
>
> |   Methods     | Cora | Citeseer | PubMed |
> |:-----------------------------|:----:|:--------:|:------:|
> | NDLS with 2*max_degree/degree | 83.2 | 71.3 | 80.1 |
> |         NDLS with LSI         | **84.6** |  **73.7** | **81.4** |
>
>
> ### 6. How certain hyperparameters were chosen:
> Both $\small k$ = 200 in **stage (1)** and $\small k$ = 40 in **stage (3)** are not hyperparameters. The only parameter of calculating LSI is $\small \epsilon$ in **Eq.(4)**, which determines the $\small K(i, \epsilon)$ in different stages. We use grid-search to select the optimal $\small \epsilon$, which can get the highest predictive accuracy on the validation set. To avoid repeatedly computing $\small I(k)$ when tuning $\small \epsilon$, we precompute $\small I(k)$ for $\small k \leq$ 200 and 40 at stage (1) and stage (3), respectively. In this way, we need to compute the corresponding $\small I(k)$ only if the tuned $\small \epsilon$ requires $\small I(k)$ for $\small k$ > 200 or 40. So the values of $k$ here are just used to improve the **efficiency of tuning** and will not affect the model performance.
>
>
> ### 7. Anonymity issue:
> The username in the local path is just a pseudonym, which will not introduce the anonymity issue.

---

> > ### Comment · Reviewer_8aSG · 2021-08-22
> > **Helpful responses**
> >
> > Thank you for the helpful responses that clarifies many issues. Glad to hear that the industrial dataset will be made public.

---

> > > ### Author Response · Authors · 2021-08-30
> > > **Response to Reviewer-2 8aSG**
> > >
> > > Thanks! We are glad to respond if you have any new questions.

---

### Official Review · Reviewer_1k7Z · 2021-07-16

**Rating:** 6
**Confidence:** 3

**Summary:**

This paper proposes a novel algorithm, node-dependent local smoothing (NDLS), which allows controlling the smoothness of every node by setting a node-specific smoothing iteration and applies to both feature smoothing and label smoothing. NDLS theoretically analyzes what influences the smoothness and gives a bound to guide how to control the extent of smoothness for different nodes. Extensive experiments on seven real-world graph datasets demonstrate that NDLS pipeline enjoys state-of-the-art performance on node classification tasks, can be combined with any GNN models, and is scalable and efficient.

**Limitations And Societal Impact:**

The potential negative social impact of the work is not presented in the paper. Although the reviewer does not foresee any potential negative impacts, the authors are encouraged to add such discussions if space permits.

**Main Review:**

----------Strengths----------

(1) The proposed NDLS algorithm is novel and generalizes some existing smoothing techniques.

(2) NDLS shows state-of-the-art performance across various datasets. Thus the algorithm is robust and generally reliable in practice.

(3) The paper is generally well written. The authors provide some theoretical analysis of the space and time complexities of the proposed NDLS pipeline. Moreover, the experimental results are summarized using some nice figures.

(4) The NDLS kernel can act as a drop-in replacement for other graph kernels and be combined with some existing models.

----------Weaknesses----------

(1) The NDLS pipeline effectively unifies and generalizes the feature and label smoothing algorithms used in GCN, SGC, APPNP, etc. The authors claim NDLS is advantageous because it allows controlling the smoothness to avoid under-smoothing or over-smoothing. However, it seems that this work does not provide deeper insight into the success of smoothing methods, especially why the control of smoothness is critical to improve the existing feature and label smoothing algorithms. In this sense, the overall contribution seems limited and a little bit incremental. The performances gain in experiments are also relatively small.

(2) Apart from SGC, SIGN, and GBP, I think it is also a good idea to compare with C&S [1] if possible. C&S uses a different smoothing procedure which is called Correct and Smooth (C&S). It is interesting to discuss whether the proposed NDLS algorithm can be combined with C&S.

----------Overall----------

Overall, I think this work is an interesting generalization of the existing smoothing algorithms for graph learning. However, it seems unclear to me why setting a node-specific smoothing iteration is crucial. Hence, I recommend weak rejection.

----------References----------

[1] Huang, Q., He, H., Singh, A., Lim, S.N. and Benson, A.R., 2020. Combining label propagation and simple models out-performs graph neural networks. arXiv preprint arXiv:2010.13993.

**Time Spent Reviewing:**

4

---

> ### Author Response · Authors · 2021-08-10
> **Response to Reviewer-1 1k7Z**
>
> We appreciate your assessment about the proposed NDLS algorithm "being novel and generalizes some existing smoothing techniques". Thanks for your constructive feedback! We believe that addressing this feedback will make our paper significantly stronger.
>
>
> ### 1. Why node-specific smoothing:
> Our paper reveals that the nodes are inherent heterogeneous across the graph; hence ***setting the same number of smoothing iterations for all nodes is suboptimal in GNNs***. To investigate this problem in a principled manner, we propose a smoothness metric and define LSI (local smoothing iterations) for each node as the number of iterations to reach the appropriate level of smoothness. The resulting LSI (shown in **Figure 1(b)**) has a heavy-tailed distribution, indicating that the control of smoothness is a **node-dependent** issue and should be treated in a node-wise and adaptive manner.
> 1. **Figure 1(a)** shows two nodes with different structures covering a dramatically different number of nodes after two smoothing iterations. Besides, **Figure 1(b)** illustrates that the speed and required iterations to approach the over-smoothing stationarity are diverse across nodes. Due to this heterogeneity, the lack of flexibility in the fixed smoothing iteration paradigm can leave the algorithm design in a **dilemma**: a small iteration number leads to insufficient aggregation and **under-smoothing** for a certain portion of nodes. However, as the smoothing iteration number increases, the majority of nodes are **over-smoothed** and indistinguishable. Both two design choices incur a significant loss of performance. To deal with the dilemma, we advocate explicitly setting **node-specific iterations** of aggregation based on LSI heterogeneity, enabling the effective treatment of long-range dependencies and avoiding of over-smoothing. It is the key reason why NDLS works better than previous fixed-iteration works like S$^2$GC [Zhu et al. 2021] and GBP [Chen et al. 2020].
> 2. To demonstrate the **node-specific requirement**, we also apply vanilla GCN with different layers (smoothing iterations) to conduct node classification on the Citeseer dataset. The figure in **appendix A.6** illustrates the ratio of correctly predicted in 50 runs of 20 randomly sampled nodes. We observe that while most of the nodes are classified correctly after 2-hop aggregation, some nodes such as nodes No.4 and 16 require more iterations of aggregation. Our work provides an in-depth understanding and efficient solution for such requirements from a new perspective of **LSI heterogeneity**, which lays a solid ground for future research developments in the field.
>
> [Zhu et al. 2021] Hao Zhu, Piotr Koniusz. [*Simple Spectral Graph Convolution.*](https://openreview.net/forum?id=CYO5T-YjWZV) ICLR 2021.
>
> [Chen et al. 2020] Ming Chen, Zhewei Wei, Bolin Ding, Yaliang Li, Ye Yuan, Xiaoyong Du, Ji-Rong Wen. [*Scalable Graph Neural Networks via Bidirectional Propagation.*](https://proceedings.neurips.cc/paper/2020/hash/a7789ef88d599b8df86bbee632b2994d-Abstract.html) NeurIPS 2020.
>
> ### 2. Comparison to C&S:
> Thanks for reminding us of the related Correct and Smooth (C&S) [Huang et al. 2020] method. We identify the following important differences between NDLS-L and C&S. Here, NDLS-L is used for comparison since it also applies a post-processing method like C&S.
> 1. ***Adaptivity to node***: C&S adopts a propagation scheme based on Personalized PageRank (PPR), which always maintains certain input information to slow down the occurrence of over-smoothing. The expected number of the smoothing iterations is controlled by the restart probability $\small \alpha$, which is **a constant for all nodes**. Therefore, C&S still falls into the routine of fixed smoothing iteration. Instead, NDLS-L employs **node-specific smoothing iterations**. We compare each method's performance (test accuracy, %) under the same data split as in the C&S paper (60%/20%/20% on three citation networks), and we find that NDLS-L **outperforms C&S**.
>     |  Methods   | Cora | Citeseer | PubMed | ogbn-papers100M |
>     |:----------|:----:|:--------:|:------:|:---------------:|
>     |  MLP+C&S   | 87.2 |   76.6   |  88.3  |      63.9       |
>     | MLP+NDLS-L | **88.1** |   **78.3**   |  **88.5**  |      **64.6**       |
>
> 2. ***Sensitivity to label rate***: During the "Correct" stage, C&S propagates uncertainties from the training data across the graph to correct the base predictions. However, the uncertainties might not be accurate when the number of training nodes is relatively small, thus even **degrading the performance**.
> To confirm the above assumption, we conduct experiments on the Cora dataset under different label rates, and the experimental results (test accuracy, %) are provided in the table below.
> As illustrated, the result of C&S drops much faster than NDLS-L's when the label rate decreases. What's more, **MLP+S (removing the "Correct" stage) outperforms MLP+C&S when the label rate is low** as expected.
>
>     | Methods    |  2%  |  5%  | 10%  | 20%  | 40%  | 60%  |
>     |:---------- |:----:|:----:|:----:|:----:|:----:|:----:|
>     | MLP+S      | 63.1 | 77.8 | 82.6 | 84.2 | 85.4 | 86.4 |
>     | MLP+C&S    | 62.8 | 76.7 | 82.8 | 84.9 | 86.4 | 87.2 |
>     | MLP+NDLS-L | **77.4** | **83.9** | **85.3** | **86.5** | **87.6** | **88.1** |
>
> Besides, NDLS is more **general** in terms of smoothing types. C&S can only smooth label predictions. Instead, NDLS can smooth **both node features and label predictions** and combine them to boost the model performance further.
>
> [Huang et al. 2020] Qian Huang, Horace He, Abhay Singh, Ser-Nam Lim, and Austin R. Benson. [*Combining Label Propagation and Simple Models out-performs Graph Neural Networks.*](https://openreview.net/pdf?id=8E1-f3VhX1o) ICLR 2020.
>
>
> ### 3. Combined with C&S:
> Very constructive feedback! The node-dependent idea in our NDLS can easily be combined with C&S. The two stages of C&S both contain a smoothing process using the personalized PageRank matrix, where a coefficient *$\small \alpha$ controls the remaining percentage of the original node feature*. Here, we can precompute the smoothed node features after the same smoothing step yet under different values of $\small \alpha$, like 0.1, 0.2, ..., 0.9. After that, we adopt the same strategy in our NDLS: for each node, we **choose the first $\alpha$ in the ascending order** that the distance from the smoothed node feature to the stationarity is less than a tuned hyperparameter, $\small \epsilon$. By this means, the smoothing process in C&S can be carried out in a **node-dependent way**.
>
> We also evaluate the performance of C&S combined with the node-dependent idea, and the following results (test accuracy, %) show that *C&S combined with NDLS consistently outperforms the original version of C&S*.
>
> |        Methods         | Cora | Citeseer | PubMed |
> |:----------------------|:----:|:--------:|:------:|
> |        MLP+C&S         | 76.7 |   70.8   |  76.5  |
> | MLP+C&S+node_dependent | **79.9** |   **71.1**   |  **78.4**  |
>
>
> ### 4. Societal impact:
> NDLS can be employed in areas where graph modeling is the foremost choice, such as citation networks, social networks, chemical compounds, transaction graphs, road networks, etc. The effectiveness of NDLS when improving the predictive performance in those areas may bring a broad range of societal benefits. For example, accurately predicting the malicious accounts on transaction networks can *help identify criminal behaviors such as stealing money and money laundering*. Prediction on road networks can *help avoid traffic overload and save people's time*. A significant benefit of NDLS is that it offers a node-dependent solution. However, NDLS faces the risk of **information leakage** in the smoothed features or labels. In this regard, we encourage researchers to understand the privacy concerns of NDLS and investigate how to mitigate the possible information leakage.

---

> > ### Comment · Reviewer_1k7Z · 2021-09-02
> > **Response to the authors**
> >
> > Thanks again for the detailed answers. I was originally concerned that the contribution of node-specific smoothing is a bit incremental, but the explanations and the added experimental results significantly mitigated my main worry. The idea is relatively simple but well-justified and can be applied to improve the "smoothing procedure" in some other algorithms. From the experimental results, the performance gain of using node-specific smoothing is evident and consistent. I would encourage the authors to add these discussions and ablation studies to the manuscript to address the impact of their work more clearly. To conclude, I am happy to increase the score to 6.

---

> ### Author Response · Authors · 2021-09-02
> **Response to Reviewer-1 1k7Z**
>
> We really appreciate your insightful and useful reviews, especially the advice of comparison and combination with C&S. In the previous response, we have explained why node-specific smoothing is crucial and added the experiments as suggested.
>
> We are very glad to respond if you have any new questions.
>
> Respectfully,
>
> Paper1237 Authors

---

> > ### Comment · Reviewer_1k7Z · 2021-09-02
> > **Thanks for your detailed feedback!**
> >
> > Thank you for your detailed feedback. Generally, feedback to my questions is satisfactory, solving some of my questions and unclear misunderstandings. I will update my review and evaluations below.

---

### Decision · Program_Chairs · 2021-09-27

**Decision:**

Accept (Spotlight)

**Comment:**

This paper proposes a node-dependent local smoothing (NDLS) algorithm to control the smoothness of every node. NDLS provides a bound to guide how to control the extent of smoothness for different nodes. Extensive experiments on seven real-world graph datasets demonstrate that NDLS pipeline enjoys state-of-the-art performance on node classification tasks, can be combined with any GNN models, and is scalable and efficient.


The proposed NDLS algorithm is novel and generalizes some existing smoothing techniques. The NDLS kernel can act as a building block to replace other graph kernels and be combined with some existing models. The authors also provide some theoretical analysis of the space and time complexities of the proposed NDLS pipeline.

The paper is very well-written and easy to follow, with intuitive figures for better understanding the concept.

NDLS shows state-of-the-art performance across various datasets. The results are nicely interpreted.  Thus the algorithm is robust and generally reliable in practice.

Therefore, we recommend accepting this paper.